# Bias Challenges in Counterfactual Data Augmentation

**S Chandra Mouli**[1]  **Yangze Zhou**[2]  **Bruno Ribeiro**[1]

[1]Department of Computer Science , Purdue University, West Lafayette, IN, USA
[2]Department of Statistics , Purdue University, West Lafayette, IN, USA

## Abstract

Deep learning models tend not to be out-of-distribution robust primarily due to their reliance on spurious features to solve the task. Counterfactual data augmentations provide a general way of (approximately) achieving representations that are counterfactual-invariant to spurious features, a requirement for out-of-distribution (OOD) robustness. In this work, we show that counterfactual data augmentations may not achieve the desired counterfactual-invariance if the augmentation is performed by a *context-guessing machine*, an abstract machine that guesses the most-likely context of a given input. We theoretically analyze the invariance imposed by such counterfactual data augmentations and describe an exemplar NLP task where counterfactual data augmentation by a context-guessing machine does not lead to robust OOD classifiers.

## 1 INTRODUCTION

Despite its tremendous success, deep learning suffers from a significant challenge of robust out-of-distribution (OOD) predictions when the test distribution is different from the training distribution, especially due to its inclination to learn spurious patterns and shortcuts to solve the task [Jo and Bengio, 2017, Geirhos et al., 2020, Poliak et al., 2018, D'Amour et al., 2020]. Invariant Risk Minimization and similar methods [Arjovsky et al., 2019, Bellot and van der Schaar, 2020, Krueger et al., 2021] propose to solve this by learning representations that are invariant across multiple environments but can be insufficient for OOD generalization without additional assumptions Ahuja et al. [2021]. Recent works have increasingly used causal language to formally define and learn non-spurious representations [Wang and Jordan, 2021, Veitch et al., 2021] in order to be robust in OOD tasks.

Veitch et al. [2021], Mouli and Ribeiro [2022] define *counterfactual invariance* to spurious features as a requirement for robust OOD predictors.

A simple way of (approximately) achieving counterfactual-invariant predictors is via counterfactual data augmentations (CDA) [Lu et al., 2020, Kaushik et al., 2019, Sauer and Geiger, 2021], where one augments the training data with inputs generated from different spurious features. This enables a predictor to learn to be invariant to these spurious features. Lu et al. [2020], Zmigrod et al. [2019], Maudslay et al. [2019] use counterfactual data augmentation to mitigate gender biases in natural language models, for example by counterfactually modifying the gendered words in the text. Kaushik et al. [2019, 2020], Teney et al. [2020] use human annotators to generate counterfactual examples by making minimal changes to a given text, although this approach may not achieve the desired robustness due to lack of diversity in augmented examples [Joshi and He, 2021]. Von Kügelgen et al. [2021] uses self-supervision and data augmentation to provably disentangle content from style in vision tasks. Other works have used pretrained models to counterfactually augment smaller datasets [Hasan and Talbert, 2021, Liu et al., 2021]. While these works propose varied ways of performing counterfactual data augmentations, the general principle remains the same: To obtain representations that are either disentangled or counterfactually-invariant to spurious features (Definition 1).

In this work, we show how counterfactual data augmentations may not achieve the desired counterfactual invariance to spurious associations if these augmentations are performed by a *context-guessing machine* (Definition 2). We define a context-guessing machine as an abstract machine (ML model, human annotator or algorithm) that infers the most-likely context of a given input $x$ before performing counterfactual modifications. We show that performing counterfactual changes with the most-likely context rather than considering all possible contexts can result in a representation that is not counterfactually invariant (Theorem 1). Our analysis suggests that one must be careful while design-

*Accepted for the 38th Conference on Uncertainty in Artificial Intelligence* (UAI 2022).

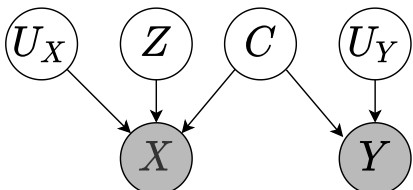

Figure 1: SCM over the observed input $X$ and the corresponding label $Y$. $X$ is obtained from two variables $C$ and $Z$ with only $C$ affecting the label $Y$ and $Z$ is spurious. $U_X$ and $U_Y$ denote the background noise variables in the SCM. The associational task is to predict $Y$ from $X$. Since $Y$ depends only on $C$, we wish to learn a representation $\Gamma : \mathcal{X} \to \mathbb{R}^d$ of the input $X$ that is counterfactually-invariant to $Z$.

ing counterfactual data augmentation methods (e.g., eliciting counterfactual examples from human annotators) to avoid the bias introduced from guessing a particular context for the given example.

## 2 COUNTERFACTUAL INVARIANCE

We begin with a brief discussion of structural causal models and the definition of counterfactual variables which will be helpful in defining counterfactual-invariant representations.

**Structural Causal Model.** A structural causal model (SCM) [Pearl, 2009, Chapter 7] describes the causal relationships between all the relevant variables and encodes the assumptions on how the observed data is generated. An SCM consists of two sets of variables: **(a)** endogenous variables, those that have a causal definition of how they are obtained from other variables, and **(b)** exogenous variables, those that are not described by the given causal model, but affect the endogenous variables. For example, consider the SCM given below

$$X = f_X(Z, C) + U_X$$
$$Y = f_Y(C) + U_Y \, ,$$

where $\{X, Y\}$ are observed endogenous variables and $\{U_X, U_Y, Z, C\}$ are unobserved exogenous variables. A (given) distribution over the exogenous variables $P(U_X, U_Y, Z, C)$ entails a distribution over the endogenous variables $P(X, Y)$. This SCM can be represented as a causal graph as shown in Figure 1. A typical learning task is to predict $Y$ from $X$; note that the task is associational, there is no causal link from $X$ to $Y$.

**Counterfactuals.** Counterfactuals describe a *what-if* question given a particular observation. For example, observing $X = x$, the counterfactual question can be "what would be the value of $X$ had $Z = z$?". We express this question using the counterfactual random variable $X(Z = z)|X = x$.

Given the complete structural causal model, the counterfactual variable $X(Z = z)|X = x$ can be computed as follows [Pearl, 2009, Chapter 7]:

1. (Abduction.) Compute $P(\mathcal{U}|X = x)$ where $\mathcal{U}$ is the set of all exogenous variables.

2. (Action.) Perform the intervention $do(Z = z)$ in the given SCM.

3. (Prediction.) Compute the distribution of $X$ in the modified SCM using the modified distribution for the exogenous variables.

We can write distribution of the counterfactual variable $X(Z = z)|X = x$ formally as:

$$P(X(Z = z) = x'|X = x) =$$
$$\int P(X = x'|do(Z = z), \mathcal{U} = \boldsymbol{u}) dP(\mathcal{U} = \boldsymbol{u}|X = x) \, .$$
$$(1)$$

The integral denotes the three steps of computing the counterfactuals: **(i)** abduction step to obtain the distribution of the exogenous variables $P(\mathcal{U}|X = x)$, **(ii)** making the desired intervention $do(Z = z)$, and **(iii)** computing the endogenous variable $X$ under intervention using the abducted distribution. An important point to remember while computing counterfactuals is that they need not only deal with individual realizations, i.e., given $X = x$, there can be a population of individuals given by the distribution $P(\mathcal{U}|X = x)$. The do-operation is then applied to all these individuals.

**Counterfactual-invariant representations.** Now we are ready to describe counterfactual-invariant representation defined in Mouli and Ribeiro [2022], Veitch et al. [2021]. Veitch et al. [2021] define the counterfactual variable using the potential outcomes notation $X(z')$, i.e., what would $X$ be had $Z = z'$ *leaving all else fixed*. While we use the notation of Mouli and Ribeiro [2022], the definitions are equivalent when $\text{supp}(Z^{\text{te}}) = \text{supp}(Z)$, as assumed throughout this work.

**Definition 1** (Counterfactual-invariant representations [Mouli and Ribeiro, 2022]). *Given any SCM with at least two variables $X$ and $Z$, a representation $\Gamma_{cf} : \mathcal{X} \to \mathbb{R}^d$, $d \geq 1$ of $X$ is counterfactual-invariant to the variable $Z$ if*

$$\Gamma_{cf}(x) = \Gamma_{cf}(X(Z = z')|X = x) \qquad (2)$$

*almost everywhere, $\forall z' \in supp(Z), \forall x \in supp(X)$, where $supp(A)$ is the support of random variable $A$.*

The counterfactual variable $X(Z = z'|X = x)$ in the RHS of Equation (2) is as defined in Equation (1). Then, Equation (2) says that $\Gamma_{cf}$ should have the same output $\Gamma_{cf}(x)$ for all values in the support of the counterfactual random variable $X(Z = z')|X = x$. Revisiting the SCM in Figure 1, we can see that an OOD robust classifier should use

representations that are counterfactual invariant to the spurious features $Z$ (as $Z$ does not affect $Y$). A common way of obtaining counterfactual invariant representations is to augment counterfactual examples $(x', y)$ for every data sample $(x, y)$ with $x' \sim P(X(Z = z')|X = x)$ for $z' \in \text{supp}(Z)$. In the next section, we look at an example of counterfactual data augmentation in the context of classifying text reviews and showcase a scenario when it does not lead to robust classifiers.

## 3 EXAMPLE: COUNTERFACTUAL DATA AUGMENTATION IN NLP

In this section, we consider an example NLP task of predicting the helpfulness of a product review while being counterfactually-invariant to the sentiment of the review which is spurious for this task [Veitch et al., 2021].

**Structural causal model for review classification.** We begin with the structural causal model that generates the text $X$ and the helpfulness label $Y$ (associated causal graph in Figure 1). In this example, $Z$ denotes the sentiment of the reviewer about the product (like or dislike), $C$ denotes the content describing the product, $U_X$ is the type of reviewer, crudely categorized as straightforward ($U_X = 1$) or sarcastic ($U_X = -1$), and $U_Y$ is label noise (assumed to be zero). Table 1 concretely defines how these variables affect the text $X$ and its helpfulness label $Y$. Since it is not feasible to describe all possible text inputs $X$, we use placeholders $[\cdots]$ to describe the type of text, while the actual content of the review may vary across the dataset. For example, $[\text{good}_1, \text{positive tone}]$ represents a particular review with good quality content and written in a positive tone. Further assume that $P(U_X = 1) = 0.9$, i.e., straightforward reviewers are a lot more likely than sarcastic ones, $P(Z = \text{like}) = 0.5$, and $P(U_Y = 0) = 1$.

A straightforward individual's sentiment affects the text in the usual way (e.g., $Z = \text{like} \implies X$ has a positive tone). On the other hand, the effect of a sarcastic individual's sentiment on $X$ is more complicated: $Z = \text{dislike} \implies X$ has a positive tone, and $Z = \text{like} \implies X$ has a neutral tone. Now, it is clear from Table 1 (also from Figure 1) that the sentiment $Z$ is spurious for the label $Y$ which only depends on $C$. However, a classifier may not learn this invariance to $Z$ automatically from training data, especially if all the possible values of $X$ are not seen during training. Thus, our goal is to obtain a representation that is counterfactually-invariant to the sentiment $Z$ which will allow us to build OOD robust classifiers. That is, we want to augment the original dataset with counterfactual data with respect to the sentiment $Z$ in order to obtain the counterfactual-invariant representation.

**Counterfactual data augmentation (CDA).** Given a text input $x = [\text{good}_1, \text{positive tone}]$, Definition 1 enforces the invariance considering all possible contexts ($Z = \text{like}, Z = \text{dislike}$), thus considering sarcastic individuals as well as straightforward ones. Considering both contexts, CDA augments $[\text{good}_1, \text{negative tone}]$ and $[\text{good}_1, \text{neutral tone}]$, thus enforcing the following invariance over the representation $\Gamma_{\text{cf}}$: $\Gamma_{\text{cf}}([\text{good}_1, \text{positive tone}]) = \Gamma_{\text{cf}}([\text{good}_1, \text{negative tone}]) = \Gamma_{\text{cf}}([\text{good}_1, \text{neutral tone}])$.

Next we show that a bias may be introduced if the counterfactual data augmentation algorithm (e.g., using humans-in-the-loop) does not consider all the possible contexts, but instead guesses the most-likely context. We will denote this type of augmentation machine as a *context-guessing machine*. As before, consider the text input $x = [\text{good}_1, \text{positive tone}]$. A context-guessing machine infers the most-likely context (i.e., maximum a posteriori estimate) of the text as $Z = \text{like}$ due to the positive tone, thus indirectly only considering straightforward individuals with $U_X = 1$. The counterfactually augmented example is $x' = [\text{good}_1, \text{negative tone}]$ with the same label $Y = \text{helpful}$ and enforces the following invariance on $\Gamma_{\text{cda}}$: $\Gamma_{\text{cda}}([\text{good}_1, \text{positive tone}]) = \Gamma_{\text{cda}}([\text{good}_1, \text{negative tone}])$. But, clearly this is not enough for counterfactual-invariance as $\Gamma_{\text{cda}}([\text{good}_1, \text{neutral tone}])$ can be arbitrarily different. Thus, a classifier that uses the representation $\Gamma_{\text{cda}}$ is not guaranteed to be robust to OOD changes to sentiment $Z$.

## 4 CDA BY A CONTEXT-GUESSING MACHINE

In this section, we analyze the counterfactual data augmentations performed by a context-guessing machine more formally. We begin with a definition of a context-guessing machine, an abstract machine (e.g., ML model, human annotator or algorithm) that guesses the most-likely context for the given input $x$, i.e., most-likely instantiation of a parent of $x$. Throughout this section, our definitions consider a single parent $Z$ of $X$, but they can be easily extended for multiple parent (context) variables.

**Definition 2** (Context-guessing machine). *Given any SCM with $Z \to X$, a context-guessing machine assumes the context of $x$ to be $z^{MAP}(x) = \text{argmax}_z P(Z = z|X = x)$, which is the maximum a posteriori (MAP) estimate of $Z$ given $X = x$.*

In Definition 3, we define counterfactual data augmentation with such a context-guessing machine which works as follows: **(a)** Given $X = x$, the context is inferred to be $Z = z^{\text{MAP}}(x)$, and **(b)** a counterfactual example is generated *conditioned on the inferred context* while preserving the label.

| $U_X$ | $Z$ | $C$ | $U_Y$ | $X$ | $Y$ |
|---|---|---|---|---|---|
| 1 | `like` | $[\text{good}_1]$ | 0 | $[\text{good}_1, \text{positive tone}]$ | `helpful` |
| 1 | `dislike` | $[\text{good}_1]$ | 0 | $[\text{good}_1, \text{negative tone}]$ | `helpful` |
| −1 | `like` | $[\text{good}_1]$ | 0 | $[\text{good}_1, \text{neutral tone}]$ | `helpful` |
| −1 | `dislike` | $[\text{good}_1]$ | 0 | $[\text{good}_1, \text{positive tone}]$ | `helpful` |
| 1 | `like` | $[\text{poor}_1]$ | 0 | $[\text{poor}_1, \text{positive tone}]$ | `not helpful` |
| 1 | `dislike` | $[\text{poor}_1]$ | 0 | $[\text{poor}_1, \text{negative tone}]$ | `not helpful` |
| −1 | `like` | $[\text{poor}_1]$ | 0 | $[\text{poor}_1, \text{neutral tone}]$ | `not helpful` |
| −1 | `dislike` | $[\text{poor}_1]$ | 0 | $[\text{poor}_1, \text{positive tone}]$ | `not helpful` |
| ⋮ | ⋮ | ⋮ | ⋮ | ⋮ | ⋮ |

Table 1: **An example where counterfactual data augmentation with a context-guessing machine does not lead to a counterfactual-invariant representation.** The table shows a succinct description of the structural causal model for a review classification task (associated causal graph in Figure 1). $X$ and $Y$ denote the observed text review and the corresponding helpfulness label respectively. The associational task is to predict $Y$ from $X$. Input $X$ is obtained as a function of two unobserved variables $C$ and $Z$. $C$ denotes the actual content describing the product and directly affects the helpfulness label $Y$. We show two possible values for $C$ denoted $[\text{good}_1]$ and $[\text{poor}_1]$ for simplicity representing particular good and poor quality contents respectively. $Z$ denotes the sentiment of the reviewer about the product. $U_X$ denotes the different types of reviewers, straightforward ($U_X = 1$) or sarcastic ($U_X = -1$). For a straightforward individual, $Z = \texttt{like}$ implies that $X$ has a positive tone, whereas for a sarcastic individual, $Z = \texttt{dislike}$ implies that $X$ has a positive tone. Finally, since $Y$ depends only on $C$ and not on the sentiment $Z$, we wish to learn a representation $\Gamma : \mathcal{X} \to \mathbb{R}^d$ of the input $X$ that is counterfactually-invariant to $Z$ using counterfactual DA. However, a context-guessing machine infers the most likely context, for example, that positive tone is from a straightforward reviewer who likes the product, and does CDA under this context. This does not result in counterfactual invariance as the alternative context for the positive tone—a sarcastic reviewer who dislikes the product—is not considered.

**Definition 3** (Guess-CDA). *Counterfactual data augmentation derived from a context-guessing machine is defined as follows: For every $(x, y)$ in the training data $\mathcal{D}$, an augmented example is $(x', y)$ where $x' \sim P(X(Z = z)|X = x, Z = z^{MAP}(x))$ for $z \in supp(Z)$.*

The variable $X(Z = z)|X = x, Z = z^{\text{MAP}}(x)$ in Definition 3 is different from the counterfactual variable of interest $X(Z = z)|X = x$. Recall that the definition of counterfactual variables in Equation (1) involved abducted distributions over the set of exogenous $\mathcal{U}$ given the evidence. The distribution of counterfactually-augmented examples in Definition 3 is given by

$$P(X(Z = z) = x'|X = x, Z = z^{\text{MAP}}(x)) =$$
$$\int P(X = x'|do(Z = z), U_X = u)$$
$$dP(U_X = u|X = x, Z = z^{\text{MAP}}(x)), \quad (3)$$

where the abducted distribution $P(U_X = u|X = x, Z = z^{\text{MAP}}(x))$ marks the only difference from Equation (1). Next, we explicitly define the invariance imposed on a representation trained over the counterfactually-augmented data of Definition 3.

**Definition 4** (Guess-CDA-invariance). *Given any SCM with $Z \to X$, the invariance imposed on a representation*

$\Gamma_{cda} : \mathcal{X} \to \mathbb{R}^d, d \geq 1$ *of $X$ by the counterfactual data augmentation from a context-guessing machine in Definition 3 is*

$$\Gamma_{cda}(x) = \Gamma_{cda}(X(Z = z)|X = x, Z = z^{MAP}(x)) \quad (4)$$

*almost everywhere, $\forall z \in supp(Z), \forall x \in supp(X)$, where $z^{MAP}(x) = \arg\max_z P(Z = z|X = x)$ is the maximum a posteriori (MAP) estimate of $Z$ given $X = x$ and $supp(A)$ is the support of random variable $A$.*

The support of the counterfactual variable in the RHS of Equation (4) can be different than the support of that in Equation (2) as illustrated by the example in Section 3. Our next theorem formalizes this notion and states that the invariance imposed by Definition 4 on $\Gamma_{cda}$ is weaker than the desired invariance of Definition 1. Hence, when performing CDA with a context-guessing machine, we are not guaranteed to obtain a counterfactually-invariant representation.

**Theorem 1** ($\Gamma_{cda}$ of Definition 4 is not counterfactually-invariant). *Given any SCM with $Z \to X$, let $\Gamma_{cda}$ denote the representation defined in Definition 4 obtained via counterfactual data augmentation from a context-guessing machine. Then, in general, $\Gamma_{cda}$ is not counterfactual-invariant according to Definition 1.*

We prove the theorem in Appendix A.1 in two steps. **(a)** First, we show that the invariance restriction imposed over $\Gamma_{\text{cda}}$ in Definition 4 is *never stronger* than that imposed over $\Gamma_{\text{cf}}$ in Equation (2) by comparing the supports of the RHS in Equations (2) and (4). That is, we show $\bigcup_z \text{supp}(X(Z = z)|X = x, Z = z^{\text{MAP}}(x)) \subseteq \bigcup_z \text{supp}(X(Z = z)|X = x)$. **(b)** Then, we show a linear SCM example where the invariance restriction of Definition 4 is *strictly weaker* than that of Definition 1. In this simple example, Definition 1 forces $\Gamma_{\text{cf}}$ to be a constant function, whereas Definition 4 allows $\Gamma_{\text{cda}}$ to take two different values based on the input $x$.

# 5 SOLUTION

The solution to the challenge described above is relatively simple. We just need to avoid guessing the most likely context. The support (or at least all likely contexts $Z$) must be present in the data augmentation procedure. That means, for instance, giving context suggestions to human annotators either by sampling from $P(Z|X)$ or by considering all the likely contexts based on $P(Z|X)$.

# 6 CONCLUSIONS

Counterfactual invariance to spurious features is a desired property for OOD robustness of predictors. A general way of approximately achieving counterfactual invariance is via counterfactual data augmentations. In this work, we studied counterfactual data augmentations performed by a context-guessing machine and showed that a representation trained on the resultant augmented data may not be counterfactual-invariant. Our analysis suggests that one must be careful while designing counterfactual data augmentation methods (e.g., eliciting counterfactual examples from human annotators) to avoid the bias introduced from guessing a particular context for the given example.

### Acknowledgements

This work was funded in part by the National Science Foundation (NSF) Awards CAREER IIS-1943364 and CCF-1918483, the Purdue Integrative Data Science Initiative, and the Wabash Heartland Innovation Network. Any opinions, findings and conclusions or recommendations expressed in this material are those of the authors and do not necessarily reflect the views of the sponsors.

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

# A APPENDIX

## A.1 PROOF OF THEOREM 1

**Theorem 1** ($\Gamma_{\text{cda}}$ of Definition 4 is not counterfactually-invariant). *Given any SCM with $Z \rightarrow X$, let $\Gamma_{cda}$ denote the representation defined in Definition 4 obtained via counterfactual data augmentation from a context-guessing machine. Then, in general, $\Gamma_{cda}$ is not counterfactual-invariant according to Definition 1.*

*Proof.* We prove the theorem in in two steps. **(a)** First, we show that the invariance restriction imposed over $\Gamma_{\text{cda}}$ in Definition 4 is *never stronger* than that imposed over $\Gamma_{\text{cf}}$ in Equation (2) by comparing the supports of the RHS in Equations (2) and (4). That is, we show $\bigcup_{z'} \text{supp}(X(Z = z')|X = x, Z = z^{\text{MAP}}(x)) \subseteq \bigcup_{z'} \text{supp}(X(Z = z')|X = x)$. **(b)** Next, we show a linear SCM example where invariance restriction of Definition 4 is *strictly weaker* than that of Definition 1.

(**a**): Consider the random variable $X(Z = z')|X = x$.

$$
\begin{aligned}
&P(X(Z = z') = x'|X = x) \\
&= \int P(X(Z = z') = x'|Z = z, U_X = u)dP(Z = z, U_X = u|X = x) \\
&= \int P(X = x'|do(Z = z'), U_X = u)dP(U_X = u|X = x) \,,
\end{aligned}
\tag{5}
$$

where the first term within the integral is rewritten using a do-expression and does not depend on $Z = z$.

Consider the random variable $X(Z = z')|X = x, Z = z^{\text{MAP}}(x)$.

$$
\begin{aligned}
&P(X(Z = z') = x'|X = x, Z = z^{\text{MAP}}(x)) \\
&= \int P(X(Z = z') = x'|Z = z, U_X = u)dP(Z = z, U_X = u|X = x, Z = z^{\text{MAP}}(x)) \\
&= \int P(X = x'|do(Z = z'), U_X = u)dP(U_X = u|X = x, Z = z^{\text{MAP}}(x)) \,,
\end{aligned}
\tag{6}
$$

where once again, we rewrite the first term using a do-expression.

Noting that $P(U_X = u|X = x) = 0 \implies P(U_X = u|X = x, Z = z^{\text{MAP}}(x)) = 0$,

$$
\begin{aligned}
& x' \notin \text{supp}(X(Z = z')|X = x) \implies P(X(Z = z') = x'|X = x) = 0 \\
&\implies \int P(X = x'|do(Z = z'), U_X = u)dP(U_X = u|X = x) = 0 && \text{(From Equation (5))} \\
&\implies \int P(X = x'|do(Z = z'), U_X = u)dP(U_X = u|X = x, Z = z^{\text{MAP}}(x)) = 0 \\
&\implies P(X(Z = z') = x'|X = x, Z = z^{\text{MAP}}(x)) = 0 && \text{(From Equation (6))} \\
&\implies x' \notin \text{supp}(X(Z = z')|X = x, Z = z^{\text{MAP}}(x)) \,.
\end{aligned}
$$

Thus, we have for all $z' \in \text{supp}(Z)$, $\text{supp}(X(Z = z')|X = x, Z = z^{\text{MAP}}(x)) \subseteq \text{supp}(X(Z = z')|X = x)$, and hence, $\bigcup_{z'} \text{supp}(X(Z = z')|X = x, Z = z^{\text{MAP}}(x)) \subseteq \bigcup_{z'} \text{supp}(X(Z = z')|X = x)$. This shows that invariance restriction over the representation $\Gamma_{\text{cda}}$ is never stronger than that over $\Gamma_{\text{cf}}$. Next, we show an example where the restriction over $\Gamma_{\text{cda}}$ is strictly weaker.

(**b**): Consider a simple SCM with $X = Z + 2U_X$ where $U_X \in \{-1, 0, 1\}$, $Z \in \{-1, 1\}$ and subsequently $X \in \{-3, -1, 1, 3\}$. Let $P(U_X = 1) = P(U_X = -1) = 0.4$ and $P(U_X = 0) = 0.2$. Also, let $P(Z = 1) = P(Z = -1) = 0.5$.

**Invariance imposed by Definition 1 on $\Gamma_{\text{cf}}$.** For each $x \in \{-3, -1, 1, 3\}$, we need to impose the condition:

$$
\Gamma_{\text{cf}}(x) = \Gamma_{\text{cf}}(X(Z = 1)|X = x) = \Gamma_{\text{cf}}(X(Z = -1)|X = x) \,.
$$

In what follows, we show the invariance imposed with $x = 1$. Consider $X(Z = 1)|X = 1$ whose distribution is given by

$$P(X(Z = 1)|X = 1) = \sum_{u \in \{-1,0,1\}} P(X|do(Z = 1), U_X = u)P(U_X = u|X = 1)$$
$$= \delta_1 \cdot P(U_X = 0|X = 1) + \delta_3 \cdot P(U_X = 1|X = 1)$$
$$= \delta_1 \cdot \frac{1}{3} + \delta_3 \cdot \frac{2}{3},$$

where $\delta_c$ is the Dirac measure at $c$. Since the support of $X(Z = 1)|X = 1$ is $\{1, 3\}$, we obtain our first invariance constraint: $\Gamma_{cf}(1) = \Gamma_{cf}(3)$. Next consider $X(Z = -1)|X = 1$

$$P(X(Z = -1)|X = 1) = \sum_{u \in \{-1,0,1\}} P(X|do(Z = -1), U_X = u)P(U_X = u|X = 1)$$
$$= \delta_{-1} \cdot P(U_X = 0|X = 1) + \delta_1 \cdot P(U_X = 1|X = 1)$$
$$= \delta_{-1} \cdot \frac{1}{3} + \delta_1 \cdot \frac{2}{3}.$$

Since the support of $X(Z = -1)|X = 1$ is $\{1, -1\}$, we obtain $\Gamma_{cf}(1) = \Gamma_{cf}(-1)$.

Repeating the above procedure for every $x \in \{-3, -1, 1, 3\}$, we obtain the following invariance for $\Gamma_{cf}$: $\Gamma_{cf}(1) = \Gamma_{cf}(-1) = \Gamma_{cf}(-3) = \Gamma_{cf}(3)$. In words, $\Gamma_{cf}$ is forced to be constant in this example.

**Invariance imposed by Definition 4 on $\Gamma_{cda}$.** For each $x \in \{-3, -1, 1, 3\}$, we need to impose the condition:

$$\Gamma_{cda}(x) = \Gamma_{cda}(X(Z = 1)|X = x, Z = z^{MAP}(x)) = \Gamma_{cda}(X(Z = -1)|X = x, z^{MAP}(x)).$$

In what follows, we show the invariance imposed with $x = 1$. First, we can obtain $z^{MAP}(1) = \text{argmax}_{z=\{-1,1\}} P(Z = z|X = 1) = -1$. Then consider $X(Z = 1)|X = 1, Z = z^{MAP}(1)$ with distribution

$$P(X(Z = 1)|X = 1, Z = -1) = \sum_{u \in \{-1,0,1\}} P(X|do(Z = 1), U_X = u)P(U_X = u|X = 1, Z = -1)$$
$$= \delta_3,$$

where once again $\delta_c$ denotes the Dirac measure at $c$. Since the support of $X(Z = 1)|X = 1, Z = -1$ is $\{3\}$, we obtain $\Gamma_{cda}(1) = \Gamma_{cda}(3)$. The support of $X(Z = -1)|X = 1, Z = -1$ is simply $\{1\}$ and only imposes the trivial constraint $\Gamma_{cda}(1) = \Gamma_{cda}(1)$.

Repeating the above procedure for every $x \in \{-3, -1, 1, 3\}$, we obtain the following invariance for $\Gamma_{cda}$: $\Gamma_{cda}(1) = \Gamma_{cda}(3)$ and $\Gamma_{cda}(-1) = \Gamma_{cf}(-3)$. Note that unlike $\Gamma_{cf}$, $\Gamma_{cda}$ is not enforced to be a constant in this example; it can be such that $\Gamma_{cda}(1) \neq \Gamma_{cda}(-1)$. Thus, there is a representation that satisfies the invariance imposed in Definition 4 but is **not** counterfactually-invariant as defined in Definition 1.

$\square$