# OpenReview forum: "Bias Challenges in Counterfactual Data Augmentation"
_auai.org/UAI/2022/Workshop/CRL — CRL@UAI 2022 Poster_

### Official Review · Reviewer_GNPU · 2022-06-29
**Nice example of subtle bias that can occur in counterfactual data augmentation**

**Rating:** 7
**Confidence:** 3

**Review:**

This is a really clear paper that explores a subtle bias that can be introduced by a "context guessing machine" (e.g. a human annotator) when they generate data augmentations. I don't have a lot to say because the bias that the authors raise seems natural and worth addressing (or at least being aware of!) in followup work. My only substantive comment is I would have found section 2 and 3 clearer if section 4 came first - it would let you explain all the definitions and what goes wrong in the context of a working example...

---

### Meta-Review · Program_Chairs · 2022-07-06

**Recommendation:** Accept (Poster)
**Confidence:** 2

**Metareview:**

The problem discussed by the authors was judged to be relevant and novel, and worthy of presentation in the context of the workshop. The authors are encouraged to take the reviewer's suggestion into consideration when preparing the camera-ready version.

---

### Decision · Program_Chairs · 2022-07-06

Accept (Poster)